# Effects of oxytocin administration and conditioned oxytocin on brain activity: An fMRI study

**Aleksandrina Skvortsova**[1,2]*, **Dieuwke S. Veldhuijzen**[1,2], **Mischa de Rover**[2,3,4], **Gustavo Pacheco-Lopez**[1,2,5], **Marian Bakermans-Kranenburg**[2,6], **Marinus van IJzendoorn**[7,8], **Niels H. Chavannes**[9], **Henriët van Middendorp**[1,2], **Andrea W. M. Evers**[1,2,10]

1 Health, Medical and Neuropsychology Unit, Faculty of Social and Behavioural Sciences, Institute of Psychology, Leiden University, Leiden, the Netherlands, 2 Leiden Institute for Brain and Cognition, Leiden, the Netherlands, 3 Clinical Psychology Unit, Faculty of Social and Behavioural Sciences, Institute of Psychology, Leiden University, Leiden, the Netherlands, 4 Department of Anesthesiology, Leiden University Medical Center, Leiden, the Netherlands, 5 Department of Health Sciences, Metropolitan Autonomous University (UAM), Lerma, Edo. Mex., Mexico, 6 Clinical Child & Family Studies, Vrije Universiteit Amsterdam, the Netherlands, 7 Department of Psychology, Education and Child Studies, Erasmus University Rotterdam, Rotterdam, the Netherlands, 8 Primary Care Unit, School of Clinical Medicine, University of Cambridge, the United Kingdom, 9 Department of Public Health and Primary Care, Leiden University Medical Center, Leiden, the Netherlands, 10 Department of Psychiatry, Leiden University Medical Center, Leiden, the Netherlands

* a.skvortsova@fsw.leidenuniv.nl

**Data Availability Statement:** The raw fMRI files cannot be shared publicly according to the regulations of Leiden University. The pre-

## Abstract

It has been demonstrated that secretion of several hormones can be classically conditioned, however, the underlying brain responses of such conditioning have never been investigated before. In this study we aimed to investigate how oxytocin administration and classically conditioned oxytocin influence brain responses. In total, 88 females were allocated to one of three groups: oxytocin administration, conditioned oxytocin, or placebo, and underwent an experiment consisting of three acquisition and three evocation days. Participants in the conditioned group received 24 IU of oxytocin together with a conditioned stimulus (CS) during three acquisition days and placebo with the CS on three evocation days. The oxytocin administration group received 24 IU of oxytocin and the placebo group received placebo during all days. On the last evocation day, fMRI scanning was performed for all participants during three tasks previously shown to be affected by oxytocin: presentation of emotional faces, crying baby sounds and heat pain. Region of interest analysis revealed that there was significantly lower activation in the right amygdala and in two clusters in the left superior temporal gyrus in the oxytocin administration group compared to the placebo group in response to observing fearful faces. The activation in the conditioned oxytocin group was in between the other two groups for these clusters but did not significantly differ from either group. No group differences were found in the other tasks. Preliminary evidence was found for brain activation of a conditioned oxytocin response; however, despite this trend in the expected direction, the conditioned group did not significantly differ from other groups. Future research should, therefore, investigate the optimal timing of conditioned endocrine responses and study whether the findings generalize to other hormones as well.

processed files are deposited on the Open Science Framework: https://osf.io/h7at3/

**Funding:** This study is funded by a European Research Council Consolidator Grant (ERC-2013-CoG-617700, granted to A. Evers). The funders had no role in study design, data collection and analysis, decision to publish, or preparation of the manuscript.

**Competing interests:** The authors have declared that no competing interests exist.

## Introduction

It has been shown that after repeated administration of medication that triggers a physiological change (unconditioned response), with an initially neutral conditioned stimulus (CS; such as, a taste or smell of the medication or the medication administration procedure), the CS alone can cause this physiological change [1]. This principle is known as pharmacological conditioning. It has been proposed that physiological responses to a CS help organisms to adapt their state in preparation for an upcoming change and in this way maintain homeostasis [2]. The principle of pharmacological conditioning has been demonstrated for various hormonal and immune parameters. For example, some evidence indicates that cortisol levels can be decreased by presenting participants with a distinctive drink previously coupled with a sumatriptan injection [3] and increased by giving a placebo injection that was previously coupled with dexamethasone [4]. Evidence of the effects of pharmacological conditioning also exists for other hormones, such as growth hormone [4] and insulin [5], and for immune parameters, such as interleukin-2 [6], natural cell killer activity [7] and histamine [8].

Despite extensive research in the field of pharmacological conditioning, no studies so far investigated neural mechanisms underlying this phenomenon. From the pain conditioning literature, it is known that conditioned analgesia decreases brain activation in pain-sensitive brain regions, such as the thalamus, insula, and dorsal anterior cingulate cortex [9]. It can be hypothesized that, similarly to the pain conditioning findings, pharmacological conditioned responses might trigger similar brain areas that are activated by the unconditioned stimulus (i.e., the medication).

In the present study, we examined for the first time the neural underpinnings of pharmacological conditioning with oxytocin. Oxytocin is a peptide hormone produced in the hypothalamus and is found to have a wide range of effects on brain activity. Its receptors are densely situated in the hypothalamus, amygdala, olfactory bulbs, and cingulate cortex [10], areas that are also associated with maternal care, social attachment and emotional processing [11]. It has been repeatedly demonstrated that exogenously administered oxytocin modifies brain responses to emotional [12, 13] and aversive visual stimuli [14], motivational tasks, involving trust [15], empathy [16], reward [17], and pain [18–20]. Particularly, oxytocin has been shown to reduce amygdala activation in response to aversive [14] and painful stimuli [20], which can be an underlying mechanism of stress reducing [21] and analgesic [22] effects of oxytocin described in previous research. Considering these positive physiological and psychological effects of oxytocin, pharmacological conditioning of oxytocin might have important clinical implications both in somatic and mental health. In this randomized placebo-controlled trial, we investigated the effects of oxytocin administration and conditioned oxytocin in comparison to a placebo-control group on brain activation in response to fMRI tasks that have previously shown to be affected by exogenous oxytocin administration: presentation of emotional faces [12], presentation of crying baby sounds [23] and thermal pain stimulation [18, 20]. We expected that the conditioned oxytocin group would demonstrate comparable brain activation patterns as the group that received exogenous oxytocin. Particularly, in response to the presentation of emotional faces, we expected that exogenous and conditioned oxytocin would reduce the activation in bilateral amygdala, and increase the activation in the insula, the occipital fusiform gyrus, and the superior temporal gyrus. We also expected that exogenous and conditioned oxytocin would decrease activation in the bilateral amygdala and increase activation in the insula and the inferior frontal gyrus pars triangularis in response to the sounds of crying babies. Finally, we expected that exogenous and conditioned oxytocin would decrease activation in the bilateral amygdala in response to pain stimulation. We also hypothesized that the

changes in brain activation triggered by conditioned oxytocin, would be smaller in magnitude than the changes cause by exogenous oxytocin administration.

## Materials and methods

### Participants

This study is part of a study on the effects of pharmacological conditioning of oxytocin effects in which in total 99 healthy female volunteers were included [24]. Of this initial sample, 88 participants took part in the MRI part of this study (11 participants did not continue with the last part due to health and planning reasons). Participants were randomly (based on a 1:1:1 ratio with a block randomization and a block size of 8) assigned to three groups: an oxytocin administration group (29 participants), a conditioned oxytocin group (29 participants), and a placebo control group (30 participants). Participants were screened for the following exclusion criteria: intake of analgesic and anti-inflammatory medication at the moment of the experiment, psychiatric, somatic, severe neurological or neurosurgical conditions that could interfere with the participant's safety or the study protocol, left-handedness, non-removable metal parts in the body, claustrophobia, (intended) pregnancy or breast feeding, and heavy use of alcohol or drugs. Only female participants were included into the study as the effects of oxytocin on brain activation have been shown to differ between the sexes [25, 26], and although this choice limits generalizability of the findings it enhances statistical power. Moreover, only participants who used oral contraceptives were included in the trial to have a better control of menstrual cycle related hormonal changes [27]. Participants were scanned in the weeks when they used oral contraceptives, not in their stop week. Participants were asked to refrain from drinking alcohol and doing intense physical exercise 24 hours before the sessions and drinking caffeinated drinks two hours before the sessions.

The study was approved by the Medical Ethical Committee of Leiden University Medical Centre (NL52683.058.15). All participants gave written informed consent to participate in the experiment and were debriefed and financially compensated afterwards.

### Sample size

The sample size was calculated with software G*Power 3. The calculation was done on the basis of a pilot experiment on conditioning of cortisol responses performed in our lab, as the design of this pilot corresponded to the design of the present study. The effect size found in the pilot experiment was d = 0.527. It was shown that 33 participants per group were necessary to obtain a power of .95 at an alpha level of a = .05. The power analysis was aimed at the question of the possibility to condition oxytocin release and not on the fMRI part of the trial.

The number of participants excluded at each step of the experiment is presented on Fig 1. One participant was not able to perform the faces task due to a technical problem with the computer. The data of one participant from the conditioned group was excluded from the analysis due to excessive head motion (frame displacement > 1 mm on 50 slices) leaving data of 86 participants that were included in the analysis of the faces task (29 participants in the oxytocin administration group, 28 participants in the oxytocin conditioned group, 29 participants in the placebo group).

Due to a technical problem with the audio system, 5 participants were not able to perform the crying baby sounds task. Additionally, data of 5 participants were excluded due to excessive head motion (frame displacement > 1 mm on 75–322 slices). Data of 78 participants in total were therefore included into the analysis of the crying baby sounds task (26 oxytocin administration group, 23 conditioned oxytocin group, 29 placebo group).

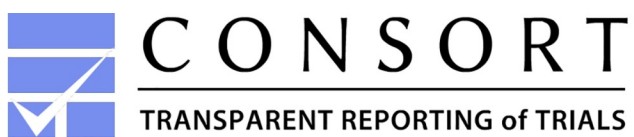

## CONSORT 2010 Flow Diagram

**Enrollment**

Assessed for eligibility (n= 114 )

Excluded (n= 10 )
- Not meeting inclusion criteria (n= 10)
- Declined to participate (n= 0)
- Other reasons (n= 0)

Randomized (n= 104 )

**Allocation**

Allocated to **Oxytocin administration** (n= 35)
- Received allocated intervention (n= 33 )
- Did not receive allocated intervention (n= 2): 1 no show, 1- participated in another pharmacological RCT
- Did not participate in MRI scanning (n=4): 2- scheduling problems; 2- no show
- 

Allocated to **Placebo control group** (n= 34)
- Received allocated intervention (n= 33)
- Did not receive allocated intervention (give reasons) (n=1): 1 – no show
- Did not participate in MRI scanning (n=3): 1- got ill on testing day; 2- scheduling problems

Allocated to **Conditioned oxytocin group** (n= 35)
- Received allocated intervention (n= 33 )
- Did not receive allocated intervention (give reasons) (n=2): 1- had to use painkillers before the first test day; 1- no show
- Did not participate in MRI scanning (n=4): 1- got ill on testing day; 3- no show

**Analysis**

**Faces task**
Analysed (n= 28 samples)
- Excluded from analysis (n= 1): excessive head motion
**Sounds task**
Analysed (n= 23)
- Did not complete the task (n=3): technical problem with the sound system
- Excluded from analysis (n= 3): excessive head motion
**Pain task**
Analysed (n= 25)
- Did not complete the task (n=4): technical problem with the termode

**Faces task**
Analysed (n=29)
- Did not complete the task (n=1): technical problem.
**Sounds task**
Analysed (n= 29)
- Excluded from analysis (n= 1): excessive head motion
**Pain task**
Analysed (n= 25)
- Did not complete the task (n=5): technical problem with the termode

**Faces task**
Analysed (n= 29)
**Sounds task**
Analysed (n=26)
- Did not complete the task (n=2): technical problem with the sound system
- Excluded from analysis (n= 1): excessive head motion
**Pain task**
Analysed (n= 24)
- Did not complete the task (n=5): technical problem with the termode

**Fig 1. CONSORT flow diagram.**

Due to technical problems with the thermode, 74 of 88 participants could take part in the pain task. Data of all of them were included in the analysis of the pain task (24 oxytocin administration group, 25 conditioned oxytocin, 25 placebo group).

## Study design

The study had a single-blind design. Participants were randomly allocated to one of the three groups and did not know whether they received the oxytocin or the placebo spray. Researchers knew which participants were included in the oxytocin administration group (due to the absence of the CS in the evocation phase). However, researchers stayed blinded regarding the conditioned oxytocin and placebo groups. The randomization was performed by the Clinical Pharmacy of Leiden University Medical Centre using block randomization. The researchers received the randomization list after the study was completed. The trial was preregistered as a clinical trial on www.trialregister.nl (number NTR5596).

## Procedures

The detailed procedures of the trial have been described elsewhere [24]. Briefly, after an initial screening, participants were randomly allocated to an oxytocin administration group, a conditioned oxytocin group, or a placebo control group. In line with previous conditioning studies [28, 29], a two-phase conditioning paradigm with an acquisition phase and an evocation phase was used. Both phases lasted for three consecutive days with a four-day break in between to allow wash-out of potential residual oxytocin effects from the previous phase. In the conditioned oxytocin group, the procedure was the following: in the acquisition phase, an association between a US (24 IU of oxytocin nasal spray) and a CS (a smell of rosewood oil) was established. Participants were administered the oxytocin spray which was immediately preceded and immediately followed by an odor of rosewood oil that was presented via a custom-made olfactometer (Wiff Online). In the evocation session, participants were administered a placebo spray paired with the same odor as in the acquisition phase. Participants in the placebo group underwent the same procedures but instead of the oxytocin spray they received a placebo spray during both phases. Participants in the oxytocin administration group received the oxytocin spray during both acquisition and evocation phases, however, they did not receive a CS during the evocation phase in order to avoid the occurrence of a conditioned response. The MRI experiment described in this study was performed on the third (last) evocation day in order to keep the conditioning context stable through the first experimental sessions.

On this third evocation day, upon arrival to the lab, participants were asked to provide a baseline saliva sample to measure their baseline oxytocin levels. Afterwards, placebo nasal spray with the odor of rosewood oil or oxytocin spray without the odor was administered, depending on group allocation. Five minutes after the spray administration, participants gave a second saliva sample and went to another lab (5-minute walk) to participate in the MRI part of the experiment.

The MRI scanning started approximately 50 minutes after the spray administration. First, an anatomical scan was performed. This was followed by several functional scans in a fixed order: the emotional faces task, the crying sounds task and finally the pain task. In total, scanning lasted for around 50 minutes.

The data were collected in the laboratory facilities of Leiden University and Leiden University Medical Centre. The data were collected between February, 2016 and August, 2017.

**Oxytocin analysis.** Oxytocin levels were measured in saliva. Each sample contained a minimum of 1.5 ml saliva that was collected with a passive drool method. Commercial ELISA kits with extraction (Enzo Life Sciences, Farmingdale, NY) were used for assaying salivary

oxytocin. Lower level of detection for oxytocin was 0.5 pg/ml after extraction. Extraction efficiency was 99%. Intra-assay coefficient of variation was 10.2%. Inter-assay coefficient of variation was 11.8%.

**Faces task.** Color photographs of males and females with four different emotional facial expressions (neutral, fearful, happy, and angry) from the Radboud Faces Database [30] were used. The pictures were grouped in blocks of 5 male and 5 female pictures of each particular valence. Each picture was presented for 1300 milliseconds with an interstimulus interval of 100 milliseconds and a block duration of 13.9 seconds. A total number of 16 blocks (4 blocks of each valence) were presented in randomized order with inter-block intervals of 10 seconds (total time of the stimulation: 7.56 minutes or 454 seconds). Stimulus presentation and response registration were controlled using E-Prime 2.0 software (Psychology Software Tools, Pittsburgh, PA).

During this task, participants were asked to focus on the screen and observe blocks of photos with faces. They were subsequently asked to rate the emotional arousal of each block on a scale from 1 (not arousing at all) to 4 (very arousing). The rating was done during the between block pause of 10 seconds. Participants could provide their ratings using button boxes placed on their thighs that were within easy reach.

**Crying baby sounds task.** The crying baby sounds task used in the current study was similar to the one that has been extensively described in previous studies [23, 31]. The crying sounds were recorded from a 2-day old child. The control sounds were digitally created identical to the crying sounds in terms of duration, intensity, spectral content, and amplitude envelope but lacking an emotional meaning [31]. The sounds were presented in 48 blocks (24 crying sounds and 24 control sounds) of 10 seconds with 6 seconds in between in randomized order. The order of the blocks was randomized within each participant. Participants were asked to focus on the sounds during the task.

**Pain task.** Pain stimuli were delivered with a standardized heat pain application device (fMRI-compatible ATS thermode attached to a Pathway device, Medoc Advanced Medical Systems, Ramat Yishai, Israel). The ATS thermode was applied to the dorsal site of the left arm of the participants when they were lying in the scanner. Before performing the functional scan, but within the scanner room, the temperature that elicited a medium pain intensity of 6 on a 0 to 10 numeric rating scale (0- no pain at all; 10- the worst pain imaginable) was identified for each participant. For this purpose, participants received a sequence of ascending temperatures with a peak temperature lasting for 5 seconds and an inter-stimulus interval of 15 seconds. Participants were asked to rate each stimulus using a 0 (no pain) -10 (worst pain imaginable) Numerical Rating Scale (NRS). The temperature that was rated as eliciting pain of 6 was used subsequently during the following functional MRI.

The pain task immediately followed the individual calibration phase and consisted of alternating 7 heat pain stimuli with a peak temperature lasting for 15 seconds, and 6 baseline stimuli of neutral to slightly warm (32 degrees Celsius) temperatures lasting between 13 and 17 seconds (on average 15 seconds) each. The purpose of the variable inter-stimulus times was to avoid anticipatory pain responses. Participants were instructed to focus on the sensation they experienced and were given the opportunity to stop the task at any moment by pressing an alarm bell. No participants pressed the alarm bell during the experiment.

## Image acquisition

The MRI data were acquired on a Philips 3T MR-system (Best, The Netherlands) in the Leiden University Medical Centre. First, a T1 weighted high resolution anatomical scan was acquired (repetition time (TR) = 9.8 milliseconds, echo time (TE) = 4.6 milliseconds, flip angle = 8˚;

voxel size 0.875 x 0.875 x 1.2; 140 slices). Functional data were acquired with echoplanar images (EPI) using a T2*-weighted gradient echo sequence (TR = 2200 milliseconds; TE = 30 milliseconds; flip angle = 80˚; voxel size 2.75 × 2.75 × 2.75 millimetres +10% slice gap, 38 transverse slices). Images were scanned parallel to the anterior–posterior commissure plane.

## Statistical analysis of demographic and psychological variables

Group differences in age, BMI, and baseline oxytocin salivary levels from the screening and three evocation days were examined using one-way analysis of variance (ANOVA).

To investigate whether there was significant conditioned oxytocin release, the conditioned oxytocin group was compared to the placebo group without adding the exogenous oxytocin group into the analysis (as extremely high oxytocin levels were expected in the exogenous oxytocin group which was decided prior to the study in the study registration protocol). The comparison was done with three (for each evocation day separately) repeated measures analyses of covariance (ANCOVA) with baseline oxytocin levels as a covariate. Next, the oxytocin administration group was added to the analyses and the three groups were compared on salivary oxytocin levels after the spray administration with repeated measures analyses of covariance in which baseline oxytocin levels served as a covariate.

To investigate whether exogenous or conditioned oxytocin had an effect on the arousal ratings given during the Faces task, arousal ratings of faces were compared between the groups with a factorial 4 (face valence: neutral, happy, angry or fearful) x 3 (group: oxytocin administration, placebo, conditioned oxytocin) ANOVA.

Finally, an ANOVA was used to compare the groups on the temperatures that elicited a pain of 6 (on an 11-point NRS) that were used during the Pain task.

## Image preprocessing and analyses

The data were pre-processed and analysed with FSL software Version 5.0.10 (FMRIB's Software Library, www.fmrib.ox.ac.uk/fsl [32]). Brain extraction from the anatomical scans was done using the Brain Extraction Tool as implemented in FSL [33]. Motion correction of functional scans was done using MCFLIRT [34]. Spatial smoothing was applied using a Gaussian kernel of full-width-at-half-maximum = 5 mm. High-pass temporal filtering was applied to the data (for faces task with high pass filter cut-off = 60 seconds; for crying baby sounds task filter = 50 seconds; for pain task filter = 90 seconds). Functional scans were registered to T1 weighted images, using Boundary-Based Registration, and then registered to an MNI-152 standard space image (Montreal Neurological Institute, Montreal, QC, Canada) using non-linear registration with a warp resolution of 10 millimetres.

The analysis consisted of three levels. The first level analysis was performed using general linear models. Blocks (for the faces task: angry, happy, fearful, neutral; for the crying baby sounds task: crying sounds, control sounds; for the pain task: painful stimuli, control stimuli) were used as predictors and convolved with a double-gamma hemodynamic response function and its temporal derivatives. Regression coefficients were estimated in FSL. For the faces task, six contrasts were estimated: angry >/< neutral, happy >/< neutral, fearful >/< neutral. For the crying baby sounds task, two contrasts were estimated: cry >/< control. For the pain task, two contrasts were estimated: pain >/< control. These first level contrasts were submitted to the second level analysis that was separately run per group. Within the second level analysis, the effects of the tasks on the brain activation were investigated in each group separately. Finally, to investigate the differences between the three groups, the third level analysis was performed. The three groups were compared with each other using analysis of variance on the contrasts. The Randomise tool of FSL was used to perform voxel-wise permutation-based

non-parametric testing and generate statistical inference for the analysis of variance. 5000 permutations per contrast were done. The statistical tests were corrected for multiple comparisons with the threshold-free cluster enhancement (TFCE). A TFCE corrected statistical threshold of $p < 0.05$ was chosen.

For exploratory purposes, these analyses were first performed on the whole brain (as in Domes and colleagues [35] and Riem and colleagues [31]). Second, a region of interest analysis was run. The regions of interest were chosen based on previous literature. Based on a recent meta-analysis on the effects of oxytocin on the brain activity in response to emotion processing tasks [26], the following regions of interest were chosen for the faces task: the bilateral amygdala, the bilateral insula, the bilateral occipital fusiform gyrus and bilateral superior temporal gyrus. For the crying baby sounds task, the bilateral amygdala, bilateral insula and the bilateral inferior frontal gyrus pars triangularis were chosen as regions of interest [31, 36]. For the pain task, left and right amygdala separately were chosen as areas of interest [20]. The masks for the regions of interest were taken from the Harvard-Oxford Cortical and Subcortical Structural Atlases (https://fsl.fmrib.ox.ac.uk/fsl/fslwiki/data/atlas-descriptions.html).

## Results

### Baseline characteristics

There were no significant differences between the three groups on age (F (2, 87) = 0.495, p = 0.611) and body mass index (F (2, 87) = 1.083, p = 0.343); the average age across the groups was 21.5 years (SD = 2.4) and mean BMI was 22.38 (SD = 2.4).

### Salivary oxytocin levels

The salivary levels of the whole sample (99 participants) during all experimental days are presented elsewhere [24]. Here we present the data of the sample of 88 participants who were included in the MRI part of the experiment. Due to clogging of the saliva samples (i.e., the saliva was thickened and could not be analyzed), 48 samples could not be analysed while 832 samples were included in the analysis. Mean salivary oxytocin levels across the groups and measurement moments and the number of analysed samples per group are presented in Table 1.There were no significant differences between the three groups on baseline salivary oxytocin levels on the screening (F (2, 80) = 1.01, p = .369), evocation day 1 (F (2, 83) = 1.47, p = 0.234), 2 (F (2, 84) = 0.37, p = 0.964) and 3 (F (2, 83) = 0.53, p = 0.588). There was a significant difference between the conditioned oxytocin and placebo groups in the levels of oxytocin

**Table 1. Mean salivary oxytocin levels (pg/ml) and standard deviations (SD) across the groups and measurement moments.**

| Test day | Measurement | Placebo group | Oxytocin administration group | Conditioned oxytocin group |
|---|---|---|---|---|
| Screening | Baseline | 12.57 (SD = 12.62, n = 30) | 9.66 (SD = 6.66, n = 25) | 15.32 (SD = 20.08, n = 26) |
| Evocation day 1 | Baseline | 16.94 (SD = 19.64, n = 30) | 12.21 (SD = 7.05, n = 26) | 11.68 (SD = 8.97, n = 28) |
| | + 5 minutes | 14.85 (SD = 7.65, n = 30) | 1912.79 (SD = 2429.66, n = 25) | 28.34 (SD = 44.63, n = 28) |
| | + 20 minutes | 13.77 (SD = 8.47, n = 30) | 1020.81 (SD = 1860.49, n = 25) | 20.17 (SD = 28.76, n = 28) |
| | + 50 minutes | 10.18 (SD = 5.1, n = 30) | 848.15 (SD = 1569.4, n = 26) | 21.17 (SD = 25.99, n = 28) |
| Evocation day 2 | Baseline | 13.23 (SD = 8, n = 30) | 13.01 (SD = 9.32, n = 27) | 13.24 (SD = 7.12, n = 28) |
| | + 5 minutes | 13.97 (SD = 6.6, n = 30) | 1727.86 (SD = 2272.05, n = 25) | 30.89 (SD = 70, n = 28) |
| | + 20 minutes | 13.14 (SD = 5.62, n = 30) | 1102.87 (SD = 1650.52, n = 25) | 17.36 (SD = 15.37, n = 28) |
| | + 50 minutes | 11.08 (SD = 5.66, n = 30) | 826.88 (SD = 1666.11, n = 26) | 10.14 (SD = 7.55, n = 28) |
| Evocation day 3 | Baseline | 16.85 (SD = 21.56, n = 30) | 14.84 (SD = 13.6, n = 26) | 12.14 (SD = 12.91, n = 28) |
| | + 5 minutes | 12.19 (SD = 7.49, n = 30) | 1719.83 (SD = 2639.64, n = 24) | 20.24 (SD = 43.93, n = 27) |

after the CS administration on the evocation day1: the conditioned oxytocin group had higher salivary oxytocin levels in comparison to the placebo group after controlling for the baseline levels (F (1, 55) = 5.98, p = 0.02). No differences between conditioned oxytocin and placebo groups were found on the salivary oxytocin level on the evocation day 2 (F (1, 55) = 1.84, p = 0.18) and evocation day 3 (F (1, 55) = 1, $p$ = 0.32). When the oxytocin administration group was added to the analysis, a significant main effect of group was found on evocation day 1 (F (2, 79) = 14.92, p < 0.001), evocation day 2 (F (2, 79) = 15.29, p < 0.001) and evocation day 3 (F (2, 80) = 11.68, p < 0.001). Post hoc Bonferroni comparison demonstrated that the oxytocin administration group had significantly higher salivary oxytocin levels than the placebo and the conditioned oxytocin group after controlling for the baseline levels during all evocation days (all p's < .001).

## Faces task

First, to examine the effects of the emotional faces' stimuli on brain activation, we looked at the second level analysis, with a specific focus on the placebo group to see the effects of the task on the brain activity regardless of the effects of oxytocin. The results of the second level analysis are presented in Table 2 (for all three groups separately) and in the Supporting Information (S1–S4 Figs). Full brain analysis in the placebo group (first column in Table 2) revealed a significant activation in two clusters in the right and left occipital fusiform gyrus, one cluster in the right superior temporal gyrus and one cluster in the right inferior temporal gyrus for the contrast fearful > neutral faces. The ROI analysis additionally showed clusters in the right amygdala, the right insula, the bilateral occipital fusiform gyrus and the bilateral superior temporal gyrus that were significantly active on the contrast fearful > neutral faces. Additionally, heightened activation in the left occipital fusiform gyrus was found on the contrast happy > neutral faces in the ROI analysis.

In the third level brain analysis we compared the three groups with each other. The whole brain analysis of variance with all three groups (oxytocin administration, conditioned oxytocin and placebo) demonstrated no significant differences in any of the three contrasts. The ROI analysis revealed that there was significantly higher activation in the right amygdala in the placebo group in comparison to the oxytocin administration group on the contrast fearful > neutral. To explore this result in detail, we plotted z-values and standard deviations of the three groups for this significant cluster (Fig 2). The z-values of the conditioned oxytocin group were in between the values of the oxytocin administration and placebo groups but did not significantly differ from either of these groups. Additionally, ROI analysis yielded a significant difference between placebo and oxytocin administration group in the left superior temporal gyrus: the activation in the oxytocin administration group was significantly lower than in the placebo group on the contrast fearful > neutral (Fig 3). The Z-values of the conditioned oxytocin group were, again, in between the values of the oxytocin administration and placebo groups but did not significantly differ from either of these groups. No significant activation was found in other regions of interest.

No differences were found between the groups in the arousal ratings given during the task (F (2, 77) = 0.15, p = .858). There was a significant difference in how arousing participants found faces of different modality (F (3, 77) = 116.85, p < .001). Bonferroni corrections demonstrated there were significant differences between all couples of modalities (all p's < .05) and that happy faces were found to be the most arousing (M = 3.00), followed by fearful faces (M = 2.75), and angry faces (M = 2.50). Neutral faces (M = 1.23) were rated as the least arousing.

Table 2. Effects of face valence across the groups (second level analysis).

| | Placebo group (n = 29) | | | | | Oxytocin administration group (n = 29) | | | | | Conditioned oxytocin group (n = 28) | | | | |
|---|---|---|---|---|---|---|---|---|---|---|---|---|---|---|---|
| | Cluster size | T max | X (mm) | Y (mm) | Z (mm) | Cluster size | T max | X (mm) | Y (mm) | Z (mm) | Cluster size | T max | X (mm) | Y (mm) | Z (mm) |
| **1. Neutral < angry** | | | | | | | | | | | | | | | |
| *ROI* left amygdala | | | | | | 2 | 6.17 | -34 | -8 | -20 | | | | | |
| *ROI* left OFG | | | | | | | | | | | 23 | 4.21 | -22 | -88 | -10 |
| **1. Neutral > angry** | | | | | | | | | | | | | | | |
| *ROI* right insula | | | | | | 20 | 4.31 | 34 | -14 | -2 | | | | | |
| **2. Neutral < happy** | | | | | | | | | | | | | | | |
| *WB* left occipital pole | | | | | | | | | | | 5 | 4.71 | 0 | -92 | -6 |
| *WB* left occipital pole superior devision | | | | | | | | | | | 50 | 5.61 | -16 | -98 | 4 |
| *WB* right occipital pole | | | | | | | | | | | 34 | 5.27 | 12 | -102 | 6 |
| | | | | | | | | | | | 16 | 6.41 | 16 | -102 | -2 |
| *ROI* left OFG extending to occipital pole | 4 | 4.36 | -18 | -96 | -4 | | | | | | 746 | 4.91 | -6 | -92 | -10 |
| **3.Neutral < fearful** | | | | | | | | | | | | | | | |
| *WB* bilateral OFG | | | | | | 4573 | 8.18 | 22 | -90 | -6 | | | | | |
| *WB* left OFG extending to lateral occipital cortex | 591 | 6.01 | -28 | -86 | -12 | | | | | | 16 | 4.79 | -36 | -84 | -10 |
| | | | | | | | | | | | 200 | 5.38 | -22 | -80 | -10 |
| *WB* right OFG extending to lateral occipital cortex | 232 | 5.15 | 28 | -88 | -10 | | | | | | 376 | 6.25 | 22 | -88 | -10 |
| *WB* right STG posterior division | 123 | 4.83 | 46 | -36 | 4 | | | | | | | | | | |
| *WB* right inferior temporal gyrus, temporooccipital part | 24 | 4.29 | 46 | -52 | -8 | | | | | | | | | | |
| *ROI* right amygdala | 16 | 3.39 | 28 | 6 | -22 | | | | | | | | | | |
| *ROI* right insula | 22 | 3.5 | 36 | 26 | 0 | | | | | | | | | | |
| *ROI* bilateral OFG | 3679 | 6.01 | -28 | -86 | -12 | 4535 | 8.18 | 22 | -90 | -6 | 869 | 5.38 | -22 | -88 | -10 |
| | | | | | | | | | | | 1088 | 6.25 | 22 | -88 | -10 |
| *ROI* left OFG | 60 | 4.76 | -46 | -12 | -12 | | | | | | | | | | |
| *ROI* right OFG | 253 | 4.83 | 46 | -36 | 4 | | | | | | | | | | |

WB- results obtained with the whole brain analysis; ROI- results obtained with the region of interest analysis; OFG- occipital fusiform gyrus; STG- superior temporal gyrus. Reported activations are corrected for multiple comparisons with the threshold-free cluster enhancement. Coordinates are reported using the Montreal Neurologic Institute space.

## Crying baby sounds task

The results of the second level analysis of the Sounds task are presented in Table 3 and in the Supporting Information (S5–S7 Figs). First, we again explored the effects of the sounds on the brain activation in the placebo group alone (first column of Table 3). The whole brain analysis showed higher activation in one large cluster in the right superior temporal gyrus with extension to the planum polare and one cluster in the left planum polare on the cry > control sounds contrasts. Subsequent ROI analysis showed that the cry > control sounds contrast caused significant activation in the right and left amygdala, the right and left insula and the left inferior frontal gyrus pars triangularis. The third level analysis with the comparison between the three groups revealed neither significant differences in the whole brain level nor in the ROI analyses.

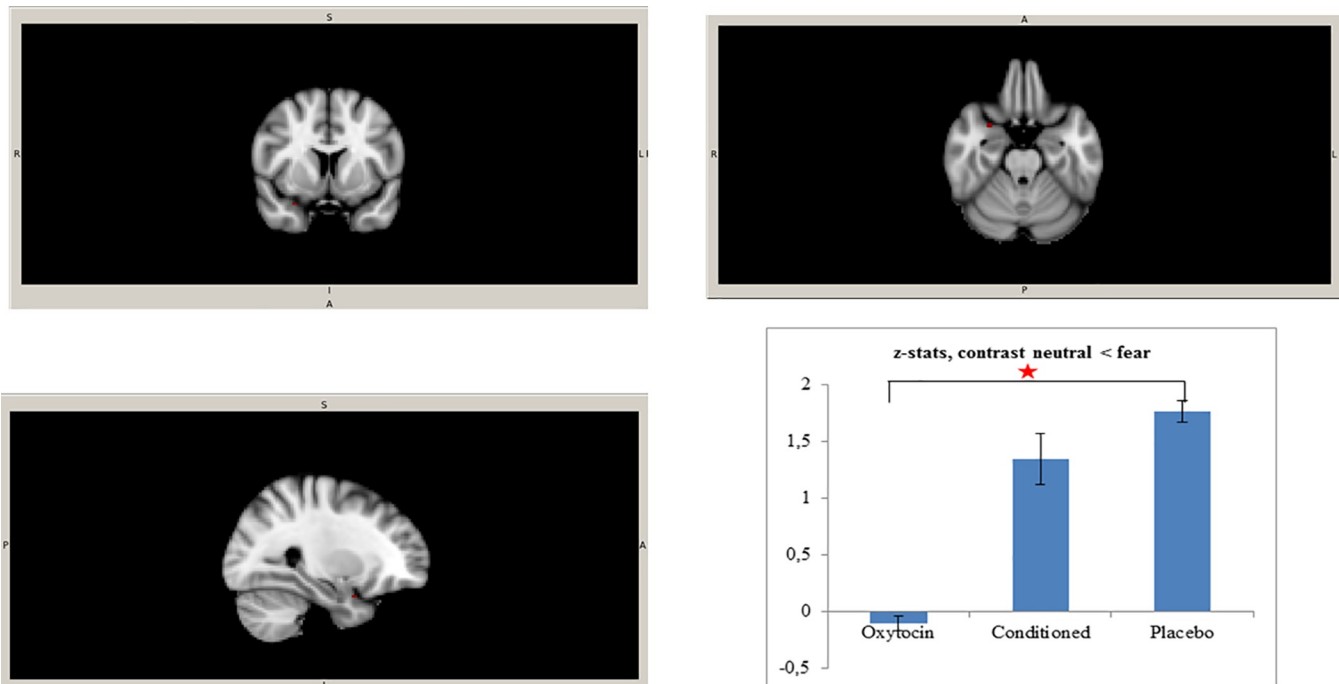

**Fig 2. Cluster in right amygdala, contrast neutral < fearful.** Cluster (28, 8, -24; t max = 3.81, cluster size = 4) with the significantly lower activation in the oxytocin group in comparison to the placebo group on the contrast neutral < fearful and Z statistics with standard deviations from this cluster.

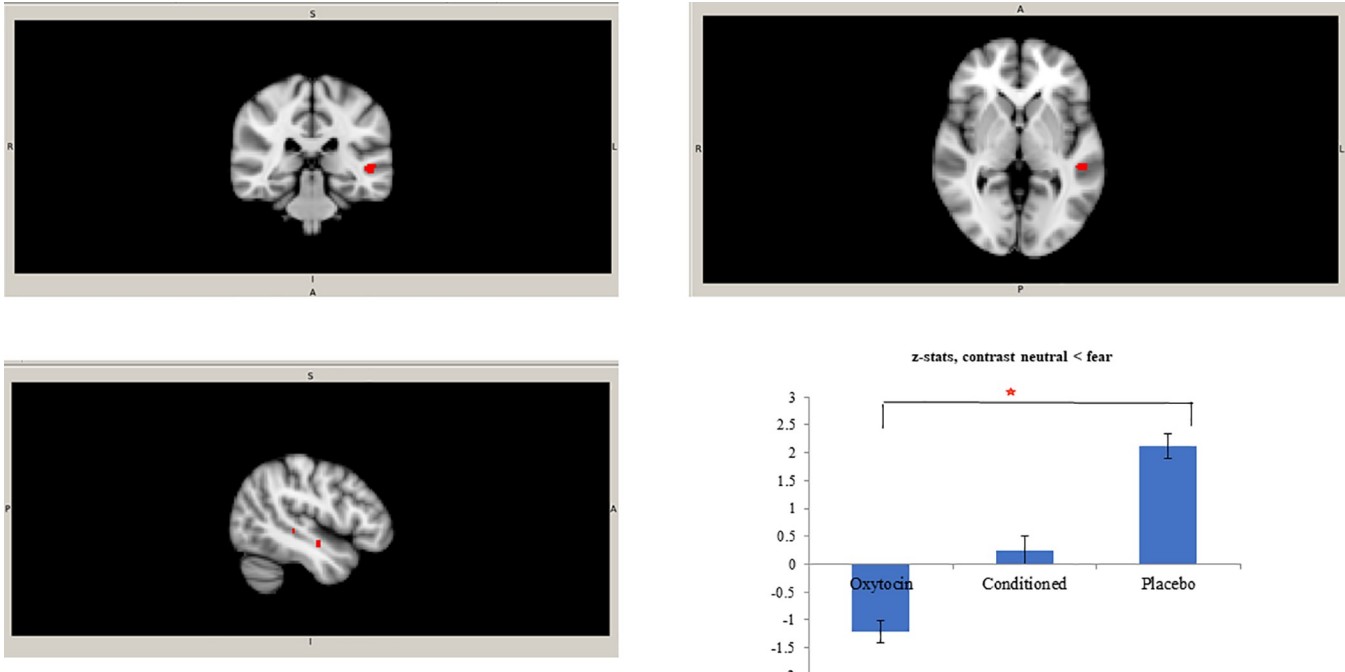

**Fig 3. Clusters in left superior temporal gyrus, contrast neutral < fearful.** Clusters in the left superior temporal gyrus (cluster 1: -50, -32, 0; t max = 4.23, cluster size = 41; cluster 2: -46, -10, -12; t max = 4.84; cluster size = 14) with the significantly lower activation in the oxytocin group in comparison to the placebo group on the contrast neutral < fearful and mean Z statistics with standard deviations from these clusters.

**Table 3. Effects of the sound valence across the groups (second level analysis).**

| | Placebo group (n = 29) | | | | | Oxytocin group (n = 26) | | | | | Conditioned group (n = 23) | | | | |
|---|---|---|---|---|---|---|---|---|---|---|---|---|---|---|---|
| | Cluster size | T max | X (mm) | Y (mm) | Z (mm) | Cluster size | T max | X (mm) | Y (mm) | Z (mm) | Cluster size | T max | X (mm) | Y (mm) | Z (mm) |
| **Cry > control** | | | | | | | | | | | | | | | |
| *WB* right STG with extension to right planum polare | 2980 | 9.75 | 58 | -10 | -2 | 1185 | 7.3 | 64 | -14 | 2 | 33 | 5.6 | 64 | -24 | 4 |
| *WB* left planum polare | 2489 | 10.1 | -54 | -4 | 2 | 1034 | 6.99 | -50 | -10 | 4 | | | | | |
| *ROI* left amygdala | 15 | 3.92 | -22 | -6 | -14 | | | | | | | | | | |
| *ROI* right amygdala | 45 | 5.09 | 32 | 8 | -20 | | | | | | | | | | |
| *ROI* left insula | 310 | 7.64 | -50 | -2 | -2 | 94 | 6.26 | -46 | -8 | 0 | | | | | |
| *ROI* right insula | 300 | 7.56 | 50 | 4 | -4 | 69 | 6.15 | 50 | -4 | -4 | | | | | |
| *ROI* left IFG, pars triangularis | 56 | 3.75 | 44 | 32 | 4 | | | | | | | | | | |

WB- results obtained with the whole brain analysis; ROI- results obtained with the region of interest analysis; STG- superior temporal gyrus; IFG = inferior frontal gyrus. Reported activations are corrected for multiple comparisons with the threshold-free cluster enhancement. Coordinates are reported using the Montreal Neurologic Institute space.

## Pain task

The results of the second level analysis of the Pain task are presented in Table 4 and in the Supporting Information (S1–S4 Figs). The effects of pain stimulation on brain activity were again first examined in the placebo group alone (first column of the Table 4). The whole brain analysis revealed significant activation in 12 clusters across the brain on the contrast pain > control and in 1 cluster on the contrast control > pain (for the details see Table 4). Significantly increased activation in both the left and right amygdala were found on the ROI analysis on the contrast pain > control. The third level analysis, comparing the three groups, revealed no significant effect of oxytocin administration or conditioning with oxytocin on brain activity neither in the whole brain analysis nor in the ROI analysis with left and right amygdala.

Finally, there were no significant differences between the three groups on the temperature that was used during the Pain task (oxytocin administration group: M = 46.2, SD = 1.1; conditioned oxytocin group: M = 46.4, SD = 1.2; placebo group: M = 45.3, SD = 1.6; F (2, 73) = 0.537, $p$ = 0.587).

## Discussion

This is the first study that investigated the effects of pharmacological conditioning with oxytocin on brain activity. We hypothesized that conditioned oxytocin responses would demonstrate patterns of brain activation similar to exogenous oxytocin administration and that both these groups would differ from the placebo group. Differences in the brain activation between the oxytocin administration and placebo groups were demonstrated in the right amygdala and in two clusters in superior temporal gyrus for the emotional faces task only, while brain activation of the conditioned oxytocin group was in between that of the oxytocin administration and the placebo group but did not significantly differ from either group.

The amygdala has been shown to play a role in negative emotion processing [37] and its activation has been found in response to threat [38]. Previous research has demonstrated that presentation of faces with fearful expression increases amygdala activation and 24 IU oxytocin has been repeatedly shown to dampen this effect [12, 13, 39]. We replicated these previous findings and furthermore showed that there is a careful indication that conditioning with oxytocin might slightly affect this activity pattern as well but to a lesser extent than exogenous

**Table 4. Effects of the pain stimulation across the groups (second level analysis).**

| | Placebo group (n = 25) | | | | | Oxytocin group (n = 23) | | | | | Conditioned group (n = 25) | | | | |
|---|---|---|---|---|---|---|---|---|---|---|---|---|---|---|---|
| | Cluster size | T max | X (mm) | Y (mm) | Z (mm) | Cluster size | T max | X (mm) | Y (mm) | Z (mm) | Cluster size | T max | X (mm) | Y (mm) | Z (mm) |
| **pain>control** | | | | | | | | | | | | | | | |
| *WB* right postcentral gyrus with extension to precentral gyrus | 10776 | 6.82 | 48 | -26 | 64 | 455 | 6.36 | 48 | -20 | 58 | 21 | 4.81 | 60 | -4 | 28 |
| *WB* left temporal occipital fusiform cortex | 1786 | 5.23 | -28 | -38 | -24 | | | | | | | | | | |
| *WB* left temporal occipital fusiform cortex with extension to left occipital fusiform gyrus | | | | | | 27763 | 8.18 | -26 | -46 | -10 | | | | | |
| *WB* right parahippocampal gyrus with extension to temporal fusiform cortex | 623 | 6.77 | 26 | -24 | -24 | | | | | | 32709 | 7.79 | 22 | -22 | -18 |
| *WB* bilateral occipital pole | 248 | 5.19 | 0 | -88 | 40 | | | | | | | | | | |
| *WB* right lateral occipital cortex | 217 | 4.89 | 44 | -68 | 26 | | | | | | | | | | |
| *WB* bilateral precuneous cortex | 162 | 4.15 | 10 | -60 | 14 | | | | | | | | | | |
| *WB* left thalamus | 118 | 4.8 | -14 | -32 | 10 | | | | | | | | | | |
| *WB* left middle temporal gyrus | 87 | 4.35 | -68 | -8 | -10 | | | | | | 25 | 4.01 | -52 | -12 | -18 |
| *WB* brain stem | 81 | 4.68 | -2 | -26 | -22 | | | | | | | | | | |
| *WB* angular gyrus | 33 | 3.45 | 48 | -54 | 30 | | | | | | | | | | |
| *WB* right cingulate gyrus | 21 | 3.57 | 8 | -50 | 6 | | | | | | | | | | |
| *WB* bilateral cuneal cortex | 15 | 3.26 | -2 | -84 | 36 | | | | | | | | | | |
| *WB* frontal pole with extension to right paracingulate gyrus | | | | | | | | | | | 1212 | 5.54 | 0 | 66 | 4 |
| *WB* left superior frontal gyrus | | | | | | | | | | | 278 | 5.12 | -24 | 22 | 42 |
| *ROI* left amygdala | 78 | 4.53 | -20 | -10 | -24 | | | | | | 18 | 5.03 | -20 | -16 | -18 |
| | 26 | 4.37 | -26 | -2 | -32 | | | | | | | | | | |
| *ROI* right amygdala | 38 | 4.55 | 14 | -4 | -24 | | | | | | 173 | 4.17 | 24 | 2 | -26 |
| **pain < control** | | | | | | | | | | | | | | | |
| *WB* right precentral gyrus | 837 | 6.52 | 46 | 10 | 26 | | | | | | | | | | |
| *WB* left precentral gyrus | | | | | | 23 | 5.55 | -58 | 6 | 4 | | | | | |
| *WB* right frontal operculum cortex to insula and central opercular cortex | | | | | | 1484 | 11 | 48 | 18 | 0 | | | | | |
| *WB* right central opercular cortex | | | | | | | | | | | 121 | 5.79 | 52 | 0 | 8 |
| *WB* right frontal pole | | | | | | 68 | 5.53 | 40 | 38 | 8 | | | | | |
| *WB* right insular cortex | | | | | | | | | | | 422 | 6.95 | 34 | 12 | 10 |

WB- results obtained with the whole brain analysis; ROI- results obtained with the region of interest analysis. Reported activations are corrected for multiple comparisons with the threshold-free cluster enhancement. Coordinates are reported using the Montreal Neurologic Institute space.

oxytocin. However, since no significant differences between the conditioned group and other two groups were found, this finding should be interpreted cautiously.

Moreover, the same fearful > neutral contrast yielded a significant difference between the oxytocin administration and placebo group in two clusters of the superior temporal gyrus (STG). The STG plays an important role in the processing of emotional stimuli and social cognition [40] and particularly processing of fearful faces [12, 41]. With this finding we thus replicated previous results showing increased STG activity in response to the presentation of the fearful faces. However, the direction of this oxytocin effect does not correspond to most previous studies. Several previous studies showed enhanced STG activity in response to emotional

and social stimuli after oxytocin administration [26, 42]. However, in our study we found that participants in the oxytocin group had lower activation in the STG on the contrast fear > neutral in comparison to the placebo group. Some studies, also found dampening effects of oxytocin on STG activation. For example, a decrease in STG activity after oxytocin administration was found in response to social rejection [43]. Also, Hech and colleagues [44] demonstrated that oxytocin reduced brain activation to social stimuli and, particularly, that individuals with higher levels of social processing exhibited oxytocin induced decrease in STG in response to social stimuli. The findings on the directionality of STG brain activity in response to oxytocin are thus mixed in the current literature. In our study, we found an increase in STG in response to fearful faces in the placebo condition and this increase was dampened by oxytocin in the oxytocin condition, corresponding to our findings in the amygdala. Again, STG activity in the conditioned oxytocin group was in between the oxytocin and placebo groups but did not significantly differ from both groups. Possibly, similar to the results of the study on social rejection [43], oxytocin inhibited the processing of negative emotions of fearful faces in our study.

The significant differences were found between oxytocin administration and placebo groups in these two areas, and at the same time the activation in the conditioned group was in between these two groups and did not significantly differ from them. This finding might be indicative of a smaller response of the conditioned group in comparison to the effect of oxytocin administration, however this should be interpreted with caution. Possibly, there was not enough power in the between-subject design of this study to find this small effect.

The sounds of a crying baby activated the auditory cortex in all groups and the amygdala and the inferior frontal gyrus, pars triangularis in the placebo group, as was expected [29]. Even though significant activation by the cry > control contrast was found in the amygdala and inferior frontal gyrus pars triangularis in the placebo group and not in the oxytocin administration and conditioned oxytocin groups on the second level analyses, the between-group comparison in the third level analysis did not reach significance. A previous study [31] found that oxytocin reduced activation in the amygdala and increased activation in the insula and the inferior frontal gyrus pars triangularis on the contrast cry > control. We could not replicate these results. Speculatively, this could be due to design differences, as Riem and colleagues [31] included twins in their sample, and performed the task 45 minutes after the oxytocin administration while our task was done approximately 60 minutes after the spray (as it followed the faces task).

The pain task activated large clusters across the brain, including primary and secondary somatosensory cortex, thalamus, cingulate gyrus and amygdala, the areas that have been repeatedly shown to be activated by acute pain [45, 46]. Importantly, several studies showed that oxytocin affects brain responses to experimentally induced pain and particularly dampens amygdala activation [18, 20, 46]. Indeed, the increased activation on the contrast pain > control in the left and right amygdala was found only in the placebo group and not in the conditioned and oxytocin administration groups in the second level analyses, but the between-group comparison was not significant. Possibly, the effects of both exogenous and conditioned oxytocin were not strong enough to be seen in the between-group comparison. The evidence about the effects of oxytocin on brain activation in response to pain, is, not conclusive. Singer and colleagues [18] found that oxytocin decreases amygdala activation in response to heat pain stimuli, however, they proposed that the effects were driven by selfish participants: effects of oxytocin on the amygdala activation were found only in selfish, but not prosocial participants. Another study by Zunhammer and colleagues [19] did not find effects of oxytocin on brain activity in response to heat pain. Speculatively, oxytocin might influence emotional or social aspects of pain perception that have not been captured by our study as, for

example, it has been shown that oxytocin enhances the pain-relieving effects of social support [47] and affects neural activity while seeing pain in others [48].

The results of both the crying baby sounds and pain task showed that the second level analyses are partially in line with the previous literature as heightened amygdala activation in response to the crying sounds and pain stimulation was found in the placebo group but not in the oxytocin administration group. However, the effects were not strong enough to be seen in the between-group comparisons. One possible explanation for this lack of significance can be the timing of the experiment. We conducted the MRI scanning on the third evocation day to avoid interference with the conditioning procedure, because it has been previously shown that presenting a distinctive additional stimulus during the conditioning might inhibit the conditioned response [49] and the whole MRI environment can be perceived to be stressful and distracting. On the third evocation day, the conditioned response in saliva had already been extinguished even though it was found on the first evocation day [24]. Since salivary oxytocin levels might not be the only indicator of the conditioned response, we still hypothesized that we could observe the conditioned response in the brain. Speculatively, if the fMRI experiment was done on the first evocation day, a stronger response in the brain might have been found. Future studies are needed to confirm this hypothesis. Moreover, the fMRI scan started approximately 50 minutes after the oxytocin and placebo administration. This time frame was chosen because the neuronal effects of exogenous oxytocin administration have been demonstrated to be the strongest around this time [50]. However, the conditioned response does not necessarily correspond to the timing of the effects of exogenous oxytocin administration. The only study on endocrine conditioning that investigated the conditioned response temporally was a study on conditioned insulin release [5] which found that the conditioned insulin release appeared around 40 minutes after the first placebo administration. However, insulin and oxytocin are distinct systems and the results from insulin conditioning cannot necessarily be directly generalized to the timing of oxytocin conditioning. As the highest conditioned oxytocin levels in saliva were found immediately after the conditioned stimulus administration on the first evocation day [24], the strongest response in the brain might possibly also immediately follow the conditioned stimulus administration and may already have decreased 50 minutes later. The fact that the only difference between the groups was found on the first task and not on the subsequent tasks supports this speculation to some extent. For future studies, it is advised that the effects of oxytocin conditioning are studied on the first evocation day, and possibly immediately following the placebo administration with repeated measurements across time to find the peak of the conditioned response.

Another possible explanation of the non-significant effects of conditioned oxytocin release on brain activity, is the difference in the magnitude of the effects between exogenous oxytocin administration and the conditioned response. Our data shows that even during evocation day 1, when the largest conditioned response was found in saliva, the oxytocin levels increased twice from the baseline, compared to a 100-time increase in the oxytocin administration group. It is unknown whether salivary oxytocin increase corresponds to the change in the brain activity in a linear way, but it can be expected that the neural effects of conditioned oxytocin release might be much smaller than the effects of oxytocin administration. However, even small natural variations of the endogenous oxytocin levels have been shown to affect brain activity, for example, in resting state [51], during massage [52], and in response to aversive stimuli [53]. Therefore, it could be expected that endogenous oxytocin release triggered by conditioning, would be strong enough to affect brain activity.

Changing hormonal levels with a behavioral manipulation can have important clinical implications especially for disorders related to dysfunction of the endocrine system. For example, it has been demonstrated that immunosuppressive treatment for renal transplant patients can be enhanced by using classically conditioned immunosuppression [28]. Nevertheless,

classically conditioned endocrine responses have not been investigated in clinical practice. The possibility to induce classically conditioned insulin release as demonstrated by Stockhorst, de Fries [5], might for example be applied for improving therapies for patients with diabetes type-2 who suffer from dysfunctional insulin release. Classical conditioning of oxytocin responses could be tested in populations with mental disorders related to emotional deficits, such as autism, schizophrenia and borderline personality disorder as oxytocin has been shown to have promising effects for treatments of these disorders [54, 55, 56]. Knowing what brain areas are involved in endocrine conditioning might be helpful for making the effects of the endocrine conditioning stronger and manipulating these effects more optimally.

To conclude, our study was the first study investigating the effects of classical conditioning with oxytocin on brain activity. We have found preliminary indications that the conditioned oxytocin response might affect the activity in the left STG and amygdala in a direction similar to exogenous oxytocin, however future studies are needed to confirm this hypothesis as no significant group difference between conditioned and other two groups was found in the current study. Moreover, these effects were not generalizable across the tasks. Future research should explore different time frames of the conditioned oxytocin brain responses and extend this paradigm to the conditioning of other hormones as comparisons to oxytocin conditioning. Unraveling the neural mechanisms of endocrine conditioning might help us to implement this potentially beneficial mechanism in clinical practice.

## Supporting information

**S1 Fig. Second level whole brain analysis in the placebo group in the faces task on the contrast neutral < fearful.**
(TIF)

**S2 Fig. Second level whole brain analysis in the oxytocin group in the faces task on the contrast neutral < fearful.**
(TIF)

**S3 Fig. Second level whole brain analysis in the conditioned group in the faces task on the contrast neutral < fearful.**
(TIF)

**S4 Fig. Second level whole brain analysis in the conditioned group in the faces task on the contrast and neutral < happy.**
(TIF)

**S5 Fig. Second level whole brain analysis in the placebo group in the crying sounds task on the contrast control < cry.**
(TIF)

**S6 Fig. Second level whole brain analysis in the oxytocin group in the crying sounds task on the contrast control < cry.**
(TIF)

**S7 Fig. Second level whole brain analysis in the conditioned group in the crying sounds task on the contrast control < cry.**
(TIF)

**S8 Fig. Second level whole brain analysis in the placebo group in the pain task on the contrast control < pain.**
(TIF)

**S9 Fig. Second level whole brain analysis in the placebo group in the pain task on the contrast control > pain.**
(TIF)

**S10 Fig. Second level whole brain analysis in the oxytocin group in the pain task on the contrast control < pain.**
(TIF)

**S11 Fig. Second level whole brain analysis in the oxytocin group in the pain task on the contrast and control > pain.**
(TIF)

**S12 Fig. Second level whole brain analysis in the conditioned group in the pain task on the contrast control < pain.**
(TIF)

**S13 Fig. Second level whole brain analysis in the conditioned group in the pain task on the contrast control > pain.**
(TIF)

## Author Contributions

**Conceptualization:** Aleksandrina Skvortsova, Dieuwke S. Veldhuijzen, Gustavo Pacheco-Lopez, Marian Bakermans-Kranenburg, Marinus van IJzendoorn, Niels H. Chavannes, Henriët van Middendorp, Andrea W. M. Evers.

**Data curation:** Aleksandrina Skvortsova, Dieuwke S. Veldhuijzen.

**Formal analysis:** Aleksandrina Skvortsova, Mischa de Rover.

**Funding acquisition:** Andrea W. M. Evers.

**Investigation:** Aleksandrina Skvortsova.

**Methodology:** Aleksandrina Skvortsova, Mischa de Rover, Henriët van Middendorp, Andrea W. M. Evers.

**Project administration:** Aleksandrina Skvortsova.

**Resources:** Niels H. Chavannes.

**Supervision:** Dieuwke S. Veldhuijzen, Mischa de Rover, Henriët van Middendorp, Andrea W. M. Evers.

**Writing – original draft:** Aleksandrina Skvortsova.

**Writing – review & editing:** Aleksandrina Skvortsova, Dieuwke S. Veldhuijzen, Gustavo Pacheco-Lopez, Marian Bakermans-Kranenburg, Marinus van IJzendoorn, Henriët van Middendorp, Andrea W. M. Evers.

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
