## [Decision Letter · Decision Letter 0]

10 Dec 2019

PONE-D-19-27410

Effects of oxytocin administration and conditioned oxytocin on brain activity: an fMRI study.

PLOS ONE

Dear Ms Skvortsova,

Thank you for submitting your manuscript to PLOS ONE. After careful consideration, we feel that it has merit but does not fully meet PLOS ONE’s publication criteria as it currently stands. Therefore, we invite you to submit a revised version of the manuscript that addresses the points raised during the review process.

Overall, the reviewers considered the paper a nice contribution to the literature, but demanded more clarity and explanation on several aspects of the paper. 

We would appreciate receiving your revised manuscript by Jan 24 2020 11:59PM. To enhance the reproducibility of your results, we recommend that if applicable you deposit your laboratory protocols in protocols.io, where a protocol can be assigned its own identifier (DOI) such that it can be cited independently in the future. For instructions see: http://journals.plos.org/plosone/s/submission-guidelines#loc-laboratory-protocols

We look forward to receiving your revised manuscript.

Kind regards,

Peter A. Bos

Academic Editor

PLOS ONE

Journal Requirements:

3. Your ethics statement must appear in the Methods section of your manuscript. If your ethics statement is written in any section besides the Methods, please move it to the Methods section and delete it from any other section. Please also ensure that your ethics statement is included in your manuscript, as the ethics section of your online submission will not be published alongside your manuscript.

Reviewers' comments:

Reviewer's Responses to Questions

**Comments to the Author**

1. Is the manuscript technically sound, and do the data support the conclusions?

Reviewer #1: Yes

Reviewer #2: Yes

Reviewer #3: Yes

2. Has the statistical analysis been performed appropriately and rigorously? 

Reviewer #1: Yes

Reviewer #2: Yes

Reviewer #3: Yes

3. Have the authors made all data underlying the findings in their manuscript fully available?

Reviewer #1: Yes

Reviewer #2: No

Reviewer #3: Yes

4. Is the manuscript presented in an intelligible fashion and written in standard English?

Reviewer #1: Yes

Reviewer #2: Yes

Reviewer #3: Yes

5. Review Comments to the Author

Reviewer #1: This is an interesting and carefully-designed study. Given the intense interest in the therapeutic applications of oxytocin for various psychiatric conditions, the possibility of a conditioned oxytocin response is intriguing and potentially of great clinical value. The authors have followed many best practices for oxytocin research, including consideration of female participants’ menstrual cycle/hormonal contraceptive use and saliva sampling to confirm no baseline oxytocin differences between groups. Another strength is the authors’ choice to use 3 distinct fMRI tasks previously shown to be influenced by exogenous oxytocin, even though they only found a significant difference in 1 task. The analysis of the fMRI data is appropriate and the authors adequately address the discrepant findings in the direction of STG activation under the oxytocin condition in the Faces task. I do hope the authors continue to pursue this line of research to obtain further evidence that a conditioned oxytocin response is indeed possible.

There are several areas of the paper that the authors should further address:

1. Abstract. The findings are stated too strongly. Specifically, the following sentence should be rephrased: “The findings carefully suggest that a conditioned response in brain activity was observed, however the conditioned group did not significantly differ from the other groups”. If there is no significant difference between the conditioned group and placebo group, you cannot say that a conditioned response was observed. Especially as this non-significant difference was only found in 1 of 3 tasks. The authors should consider describing this as preliminary evidence.

2. Salivary analysis / Table 1

The author states that 48 saliva samples could not be analyzed because they were "clogged" (p 11). It is not clear what this means. It is also relevant what groups these samples belong to, and for this reason Table 1 should include sample sizes. In Table 1, I assume the values in parentheses ( ) are the standard deviations? This is not stated. While I understand that the details of the salivary hormone analysis are presented elsewhere, given on-going concerns about salivary oxytocin measurements,the authors should briefly described the method of analysis (ie what kit was used, whether there was an extraction step).

3 Discussion (page 30)

Page 30, line 409: “On the third evocation day, the conditioned response in saliva had already been extinguished”.

The data in Table 1 does not appear to support this (Day 3, +5 min, Placebo Group 12.19 pg/ml vs Conditioned Oxytocin Group 61.23 pg/ml); In fact, the oxytocin level at +5 min in the Conditioned Oxytocin Group is 2-fold greater on Evocation Day 3 vs Evocation Day 1.

The data in Table 1 also appears to conflict with this statement “As the highest conditioned oxytocin levels in saliva were found immediately after the conditioned stimulus administration on the first evocation day” (p 31, line 422). As this statement is the basis for the further suggestion that at a greater difference in brain activation may have been observed if scanning was done on Day 1, this should be clarified.

4. Given that any conditioned oxytocin release would be much smaller than the supraphysiological dose (24 IU) of intranasal oxytocin (which is supported by your salivary hormone data), how much of an effect would you expect more naturalistic oxytocin release to have on fMRI brain activation? This might be worth discussing, as it could help explain the non-significant differences between the conditioned oxytocin group and placebo group on fMRI tasks.

5. While the overall quality of the writing is acceptable, there are small errors throughout. The authors should carefully proofread their article once more. Errors I noted include:

p 7, line 153 “data was”

p 9,line 204-205 “the oxytocin salivary …”

p 10, line 246, “regions of interests”

p 13, line 292 “conditioned oxytocin [group]”

p 21, line 324 “the table 4|”

Reviewer #2: This is an interesting, elegant and well-conducted study and a well-written manuscript. It involves an ambitious and unique oxytocin conditioning experiment, performed in three relatively large samples of adult females, implying three well-established fMRI experiments.

I definitely recommend the study for publication. I do have a number of relatively minor suggestions for further improving the report.

- Page 4, end of Introduction: It would be helpful for the reader if you could already give a brief preview on the expected patterns of major activation for each of the three fMRI paradigms, and the particular hypotheses concerning the impact of exogenous/conditioned oxytocin.

- Page 4, Participants: What is the rationale for exclusively including female participants?

- Page 4, line 92-94: “Only participants who used oral contraceptives during active use weeks …’ � unclear phrasing; please rephrase

- Page 9, Preprocessing, line 221: Unclear why the first level analysis was performed in native space? + how these data have than be brought to MNI space for the second- and third-level analysis?

- Page 10, line 238-245, concerning the choice for ROIs: For the FACES task, in addition to more general emotion processing ROIs the authors also include “modality specific” ROIs, such as fusiform gyrus and superior temporal gyrus. However, for the CRYING task and for the PAIN task, the authors do not include these modality specific ROIs, i.e. auditory processing and somatosensory processing, respectively. I think it would have been more logical if they would also have performed an ROI analysis in primary and secondary auditory regions for the CRYING tasks, and in primary and secondary somatosensory regions for the PAIN task. This more focused approach might possibly reveal additional group differences for these both tasks, as already suggested by the reported cluster characteristics drawn from the whole-brain analyses in Tables 3 and 4.

- Results FACES task: Could the authors also report the behavioural data for these tasks? Were there any group differences in perceived arousal of the emotional faces?

- Page 14, Table 2 (+ also Table 3-4): Please, verify the textual presentation in this Table and in the Legend, as it is not optimal (sometimes capital letters, sometimes not). WF � WB?

- P. 30 line 410: typo “be”

- Generally, for the three fMRI paradigms: It would be informative to have whole brain plots/figures of the most informative contrasts (i.e. fear>neutral; cry>sound; pain>control); at least for the Placebo group, but preferably for the three groups separately.

Reviewer #3: This study examines the neural underpinnings of pharmacological conditioning with oxytocin. The authors tested intranasal and conditioned oxytocin effects on brain activity in response to three fMRI tasks that has been previously been shown to be affected by intranasal oxytocin administration. This study is very interesting and novel in that there is a lack of fMRI research on conditioned oxytocin effects. However, I do have some questions/comments and concerns with the way the results are presented.

Introduction:

The rationale for using a pain stimulation task to examine neural underpinnings of oxytocin conditioning is unclear, because there is no strong evidence that oxytocin affects pain sensitivity. In the introduction, the authors refer to a thermal pain stimulation task (line 79) that has been shown to be affected by intranasal oxytocin. However, in the discussion they mention that fMRI findings on oxytocin and pain perception/sensitivity are inconsistent. It is unclear why the authors expect to find effects of conditioned oxytocin during this task. Please provide a clear rationale.

Methods/results:

-The description of the power analysis is unclear. Please provide more details.

-Description of statistical analyses (line 203-207) is unclear. Please reframe.

-Were there any group differences in baseline OT before the acquisition phase?

-Grammatical error line 261 – 262: “was found” should be “were found”.

-Were there any effects of conditioned oxytocin or intranasal oxytocin on the stimulus ratings (faces and cry sounds)?

Discussion:

The authors should be more careful with interpreting insignificant results throughout the manuscript. E.g. line 444-447 � the authors mention that “effects remained small”, but effects were non-significant.

The authors found that intranasal oxytocin reduced STG activity, whereas previous studies point to enhanced STG activity. The authors should explain this discrepancy more clearly and discuss how their findings relate to previous research.

In the discussion, the authors could elaborate more on differential effects of exogenous oxytocin and manipulated endogenous oxytocin on brain activity. The increase in oxytocin induced by the conditioning paradigm is very small compared to the enormous increase induced by intranasal administration (Table 1). The authors do not find significant differences between the placebo group and conditioned oxytocin group. I wonder whether this may be due to the small increase in oxytocin, which may be insufficient to result in detectable brain activity changes.

6. PLOS authors have the option to publish the peer review history of their article (what does this mean?). If published, this will include your full peer review and any attached files.

Reviewer #1: No

Reviewer #2: Yes: Bart Boets

Reviewer #3: No

---

## [Author Response · Author response to Decision Letter 0]

23 Jan 2020

January 23, 2020

Leiden, the Netherlands

Dear Prof. Dr. Bos,

We would like to thank you for the opportunity to submit a revised version of our manuscript, “Effects of oxytocin administration and conditioned oxytocin on brain activity: an fMRI study” for publication in PlosOne. We have revised the original manuscript to address the comments and concerns raised by the reviewers. The point-to-point responses to each comment are given below and changes in the manuscript are marked with Track Changes. We hope that with this revision we satisfactory addressed all issues. 

Sincerely, on behalf of all co-authors, 

Aleksandrina Skvortsova, PhD

Corresponding author

Journal Requirements:

Response: We have renamed the files.

Response: The data will be uploaded to the platform DataVerse.nl according to the regulations of Leiden University. We will provide you with the link as soon as the data is online. 

3. Your ethics statement must appear in the Methods section of your manuscript. If your ethics statement is written in any section besides the Methods, please move it to the Methods section and delete it from any other section. Please also ensure that your ethics statement is included in your manuscript, as the ethics section of your online submission will not be published alongside your manuscript.

Response: We thank the editorial office for this remark. We have added the information regarding the ethical approval to the Methods section on page 5. Now it reads as follows:

The study was approved by the Medical Ethical Committee of Leiden University Medical Centre (NL52683.058.15). All participants gave written informed consent to participate in the experiment and were debriefed and financially compensated afterwards.

Response: We have included captions for the supporting information files and updated the in-text citations.

Reviewers' comments:

Reviewer's Responses to Questions

Comments to the Author

1. Is the manuscript technically sound, and do the data support the conclusions?

Reviewer #1: Yes

Reviewer #2: Yes

Reviewer #3: Yes

2. Has the statistical analysis been performed appropriately and rigorously?

Reviewer #1: Yes

Reviewer #2: Yes

Reviewer #3: Yes

3. Have the authors made all data underlying the findings in their manuscript fully available?

Reviewer #1: Yes

Reviewer #2: No

Response: The data will be made available on DataVerse.nl

Reviewer #3: Yes

4. Is the manuscript presented in an intelligible fashion and written in standard English?

Reviewer #1: Yes

Reviewer #2: Yes

Reviewer #3: Yes

5. Review Comments to the Author

Reviewer #1: This is an interesting and carefully-designed study. Given the intense interest in the therapeutic applications of oxytocin for various psychiatric conditions, the possibility of a conditioned oxytocin response is intriguing and potentially of great clinical value. The authors have followed many best practices for oxytocin research, including consideration of female participants’ menstrual cycle/hormonal contraceptive use and saliva sampling to confirm no baseline oxytocin differences between groups. Another strength is the authors’ choice to use 3 distinct fMRI tasks previously shown to be influenced by exogenous oxytocin, even though they only found a significant difference in 1 task. The analysis of the fMRI data is appropriate and the authors adequately address the discrepant findings in the direction of STG activation under the oxytocin condition in the Faces task. I do hope the authors continue to pursue this line of research to obtain further evidence that a conditioned oxytocin response is indeed possible.

Response: We thank the reviewer for the supportive words and the interest in our study. 

There are several areas of the paper that the authors should further address:

1. Abstract. The findings are stated too strongly. Specifically, the following sentence should be rephrased: “The findings carefully suggest that a conditioned response in brain activity was observed, however the conditioned group did not significantly differ from the other groups”. If there is no significant difference between the conditioned group and placebo group, you cannot say that a conditioned response was observed. Especially as this non-significant difference was only found in 1 of 3 tasks. The authors should consider describing this as preliminary evidence.

Response: We thank the reviewer for this important comment. Indeed, we do not want to overstate our results in any way. We have adjusted the abstract and now it reads as follows:

Preliminary evidence was found for brain activation of a conditioned oxytocin response; however, despite this trend in the expected direction, conditioned group did not significantly differ from other groups. Future research should, therefore, investigate the optimal timing of conditioned endocrine responses and study whether the findings generalize to other hormones as well.

2. Salivary analysis / Table 1

The author states that 48 saliva samples could not be analyzed because they were "clogged" (p 11). It is not clear what this means. It is also relevant what groups these samples belong to, and for this reason Table 1 should include sample sizes. In Table 1, I assume the values in parentheses ( ) are the standard deviations? This is not stated. While I understand that the details of the salivary hormone analysis are presented elsewhere, given on-going concerns about salivary oxytocin measurements,the authors should briefly described the method of analysis (ie what kit was used, whether there was an extraction step).

Response: We thank the reviewer for these remarks. Indeed, 48 samples (5% of the total number) could not be analyzed due to clogging of the samples. It means that the saliva was thickened and could not be analyzed; this explanation is now added to the manuscript on page 12:

Due to clogging of the saliva samples (i.e., the saliva was thickened and could not be analyzed), 48 samples could not be analysed while 832 samples were included in the analysis. 

We have added the number of the analyzed samples per group and per measurement moment in Table 1. We have also specified that the number in parentheses in Table 1 is indeed the standard deviation. Moreover, we have added a paragraph describing the method of oxytocin analysis to page 7. It reads as follows:

Oxytocin analysis 

Oxytocin levels were measured in saliva. Each sample contained a minimum of 1.5 ml saliva that was collected with a passive drool method. Commercial ELISA kits with extraction (Enzo Life Sciences, Farmingdale, NY) were used for assaying salivary oxytocin. Lower level of detection for oxytocin was 0.5 pg/ml after extraction. Extraction efficiency was 99%. Intra-assay coefficient of variation was 10.2%. Inter-assay coefficient of variation was 11.8%.

3 Discussion (page 30)

Page 30, line 409: “On the third evocation day, the conditioned response in saliva had already been extinguished”.

The data in Table 1 does not appear to support this (Day 3, +5 min, Placebo Group 12.19 pg/ml vs Conditioned Oxytocin Group 61.23 pg/ml); In fact, the oxytocin level at +5 min in the Conditioned Oxytocin Group is 2-fold greater on Evocation Day 3 vs Evocation Day 1.

The data in Table 1 also appears to conflict with this statement “As the highest conditioned oxytocin levels in saliva were found immediately after the conditioned stimulus administration on the first evocation day” (p 31, line 422). As this statement is the basis for the further suggestion that at a greater difference in brain activation may have been observed if scanning was done on Day 1, this should be clarified.

Response: We thank the reviewer for noticing this inconsistency. There was indeed a mistake in Table 1. The oxytocin level in the conditioned group on the day 3, +5 minutes was 20.24 pg/ml and not 61.23 pg/ml. We have made the correction in Table 1 and double checked all other values. This number is supported by our statistical analysis that demonstrated that there were no differences between the conditioned and the placebo groups on this measurement point. 

4. Given that any conditioned oxytocin release would be much smaller than the supraphysiological dose (24 IU) of intranasal oxytocin (which is supported by your salivary hormone data), how much of an effect would you expect more naturalistic oxytocin release to have on fMRI brain activation? This might be worth discussing, as it could help explain the non-significant differences between the conditioned oxytocin group and placebo group on fMRI tasks.

Response: We thank the review for this very interesting remark. Indeed, the conditioned response found in saliva was considerably lower than the increase in salivary oxytocin levels after the exogenous oxytocin administration. However, we do not know to what extent the increase in salivary oxytocin levels reflects the conditioned response in the brain activity. Moreover, several studies have found that natural variations in endogenous oxytocin levels predict brain activity. We have added the discussion of these issues to the discussion on page 33. Now it reads as follows:

Another possible explanation of the non-significant effects of conditioned oxytocin release on brain activity, is the difference in the magnitude of the effects between exogenous oxytocin administration and the conditioned response. Our data shows that even during evocation day 1, when the largest conditioned response was found in saliva, the oxytocin levels increased twice from the baseline, compared to a 100-time increase in the oxytocin administration group. It is unknown whether salivary oxytocin increase corresponds to the change in the brain activity in a linear way, but it can be expected that the neural effects of conditioned oxytocin release might be much smaller than the effects of oxytocin administration. However, even small natural variations of the endogenous oxytocin levels have been shown to affect brain activity, for example, in resting state [50], during massage [51], and in response to aversive stimuli [52]. Therefore, it could be expected that endogenous oxytocin release triggered by conditioning, would be strong enough to affect brain activity.

5. While the overall quality of the writing is acceptable, there are small errors throughout. The authors should carefully proofread their article once more. Errors I noted include:

p 7, line 153 “data was”

p 9,line 204-205 “the oxytocin salivary …”

p 10, line 246, “regions of interests”

p 13, line 292 “conditioned oxytocin [group]”

p 21, line 324 “the table 4|”

Response: We thank the reviewer for mentioning these errors. We have adjusted them and proofread the whole manuscript. 

Reviewer #2: This is an interesting, elegant and well-conducted study and a well-written manuscript. It involves an ambitious and unique oxytocin conditioning experiment, performed in three relatively large samples of adult females, implying three well-established fMRI experiments.

I definitely recommend the study for publication. I do have a number of relatively minor suggestions for further improving the report.

Response: We thank the reviewer for the supporting words about our manuscript.

- Page 4, end of Introduction: It would be helpful for the reader if you could already give a brief preview on the expected patterns of major activation for each of the three fMRI paradigms, and the particular hypotheses concerning the impact of exogenous/conditioned oxytocin.

Response: We thank the reviewer for this advice. We have added hypotheses regarding the expected pattern of brain activation for all three tasks to the page 4:

We expected that the conditioned oxytocin group would demonstrate comparable brain activation patterns as the group that received exogenous oxytocin. Particularly, in response to the presentation of emotional faces, we expected that exogenous and conditioned oxytocin would reduce the activation in the bilateral amygdala, and increase the activation in the insula, the occipital fusiform gyrus, and the superior temporal gyrus. We also expected that exogenous and conditioned oxytocin would decrease activation in the bilateral amygdala and increase activation in the insula and the inferior frontal gyrus pars triangularis in response to the sounds of crying babies. Finally, we expected that exogenous and conditioned oxytocin would decrease activation in the bilateral amygdala in response to pain stimulation. We also hypothesized that the changes in brain activation triggered by conditioned oxytocin, would be smaller in magnitude than the changes cause by exogenous oxytocin administration.

- Page 4, Participants: What is the rationale for exclusively including female participants?

Response: It has been repeatedly shown that sex is an important moderator of the effects of oxytocin on brain activity (Rilling et al., Psychoneuroendocrinology, 2014; Grace et al., Psychoneuroendocrinology, 2018). Therefore, we decided to limit our sample to only female participants to avoid a possible confounding effect of sex differences. Moreover, this choice allowed us to enhance statistical power as we did not have to take into account a possible interaction term with sex. We have added this explanation to the manuscript on page 5:

Only female participants were included into the study as the effects of oxytocin on brain activation have been shown to differ between the sexes (25, 26), and although this choice limits generalizability of the findings it enhances statistical power.

- Page 4, line 92-94: “Only participants who used oral contraceptives during active use weeks …’ � unclear phrasing; please rephrase

Response: We have adjusted the sentence. Now it reads as follows (page 5):

Only participants who used oral contraceptives were included in the trial to have a better control of menstrual cycle related hormonal changes [27]. Participants were scanned in the weeks when they used oral contraceptives, not in their stop week.

- Page 9, Preprocessing, line 221: Unclear why the first level analysis was performed in native space? + how these data have than be brought to MNI space for the second- and third-level analysis?

Response: We thank the reviewer for this important question. The first level analysis was indeed not performed in the native space. Before the first level analysis, all functional scans were registered to T1 weighted images, using Boundary-Based Registration, and then registered to an MNI-152 standard space image; this procedure is described on page 10.

- Page 10, line 238-245, concerning the choice for ROIs: For the FACES task, in addition to more general emotion processing ROIs the authors also include “modality specific” ROIs, such as fusiform gyrus and superior temporal gyrus. However, for the CRYING task and for the PAIN task, the authors do not include these modality specific ROIs, i.e. auditory processing and somatosensory processing, respectively. I think it would have been more logical if they would also have performed an ROI analysis in primary and secondary auditory regions for the CRYING tasks, and in primary and secondary somatosensory regions for the PAIN task. This more focused approach might possibly reveal additional group differences for these both tasks, as already suggested by the reported cluster characteristics drawn from the whole-brain analyses in Tables 3 and 4.

Response: We thank the reviewer for this important remark. Choosing the ROIs for our analyses, we decided to choose a confirmative approach and focus on the areas that specifically have been shown in previous research to be affected by oxytocin administration during these specific tasks. For the task with presentation of emotional faces, we included the ROIs that have been shown to be affected by oxytocin in previous studies using this task, and these areas were amygdala, insula, fusiform gyrus and superior temporal gyrus (Domes et al., Psychoneuroendocrinology, 2010; Domes at al., Biological psychiatry, 2007; meta-analysis: Grace et al., Psychoneuroendocrinology, 2018). Therefore, by choosing fusiform gyrus and superior temporal gyrus we were not aiming to select “modality specific” ROIs but rather the ROIs that were already found in the previous research. We used the same principle in selecting ROIs for the crying sounds and pain tasks. The previous research on these two tasks did not find effects of oxytocin on the auditory cortex and somatosensory cortex respectively, and therefore, we did not include these regions in our ROI analysis. This approach we took by selecting ROIs based on previous findings is described on page 11. 

- Results FACES task: Could the authors also report the behavioural data for these tasks? Were there any group differences in perceived arousal of the emotional faces?

Response: We thank the reviewer for this important question. There were no differences found between the groups in the arousal ratings. We have added the description of the analysis and results of this task to the methods on page 10 and results on page 15. Now it reads as follows:

Page10:

To investigate whether exogenous or conditioned oxytocin had an effect on the arousal ratings given during the Faces task, arousal ratings of faces were compared between the groups with a factorial 4 (face valence: neutral, happy, angry or fearful) x 3 (group: oxytocin administration, placebo, conditioned oxytocin) ANOVA.

Page 15:

No differences were found between the groups in the arousal ratings given during the task (F (2, 77) = 0.15, p = .858). There was a significant difference in how arousing participants found faces of different modality (F (3, 77) = 116.85, p < .001). Bonferroni corrections demonstrated there were significant differences between all couples of modalities (all p’s < .05) and that happy faces were found to be the most arousing (M = 3.00), followed by fearful faces (M = 2.75), and angry faces (M = 2.50). Neutral faces (M = 1.23) were rated as the least arousing.

- Page 14, Table 2 (+ also Table 3-4): Please, verify the textual presentation in this Table and in the Legend, as it is not optimal (sometimes capital letters, sometimes not). WF � WB?

Response: We thank the reviewer for these remarks. We have adjusted the Tables.

- P. 30 line 410: typo “be”

Response: We thank the reviewer for noticing this. We have corrected the typo. 

- Generally, for the three fMRI paradigms: It would be informative to have whole brain plots/figures of the most informative contrasts (i.e. fear>neutral; cry>sound; pain>control); at least for the Placebo group, but preferably for the three groups separately.

Response: We thank the reviewer for this advice. We have added figures with the results of the whole brain second level analysis to the Supporting Information. 

Reviewer #3: This study examines the neural underpinnings of pharmacological conditioning with oxytocin. The authors tested intranasal and conditioned oxytocin effects on brain activity in response to three fMRI tasks that has been previously been shown to be affected by intranasal oxytocin administration. This study is very interesting and novel in that there is a lack of fMRI research on conditioned oxytocin effects. However, I do have some questions/comments and concerns with the way the results are presented.

Response: We thank the reviewer for expressing interest in our study. 

Introduction:

The rationale for using a pain stimulation task to examine neural underpinnings of oxytocin conditioning is unclear, because there is no strong evidence that oxytocin affects pain sensitivity. In the introduction, the authors refer to a thermal pain stimulation task (line 79) that has been shown to be affected by intranasal oxytocin. However, in the discussion they mention that fMRI findings on oxytocin and pain perception/sensitivity are inconsistent. It is unclear why the authors expect to find effects of conditioned oxytocin during this task. Please provide a clear rationale.

Response: We thank the reviewer for raising this important question. Indeed, the effects of oxytocin on pain perception remain unclear: some studies found that oxytocin decreases pain sensitivity, while others failed to confirm these results (see the systematic review: Rash et al., 2014, The Clinical journal of pain). However, we have chosen to include the heat pain task because a body of evidence points at the modulatory effects of oxytocin on the brain activation in response to pain: Zunhammer et al., 2015, Psychosomatic Medicine; Singer et al 2008, Emotion; Kreuder et al., 2019, Human Brain Mapping, for seeing pain in others: Bos et al., 2005, Neuroimage. One study that did not find modulatory effects of oxytocin on brain activity in response to pain, is the study by Zunhammer and colleagues, 2016, Scientific reports. On the basis, of the existing literature, we concluded that pain stimulation, might be a relevant task for investigating the effects of exogenous and conditioned oxytocin; however, our hypothesis was not confirmed. 

We have made small modifications in the paragraph where we discuss the results of the pain task to make the rationale clearer (page 31):

The pain task activated large clusters across the brain, including primary and secondary somatosensory cortex, thalamus, cingulate gyrus and amygdala, the areas that have been repeatedly shown to be activated by acute pain [ 44, 45]. Importantly, several studies showed that oxytocin affects brain responses to experimentally induced pain and particularly dampens amygdala activation [18, 20, 46]. Indeed, the increased activation on the contrast pain > control in the left and right amygdala was found only in the placebo group and not in the conditioned and oxytocin administration groups in the second level analyses, but the between-group comparison was not significant. Possibly, the effects of both exogenous and conditioned oxytocin were not strong enough to be seen in the between-group comparison. The evidence about the effects of oxytocin on brain activation in response to pain, is however not conclusive. Singer and colleagues [18] found that oxytocin decreases amygdala activation in response to heat pain stimuli, however, they proposed that the effects were driven by selfish participants: effects of oxytocin on the amygdala activation were found only in selfish, but not prosocial participants. Another study by Zunhammer and colleagues [19] did not find effects of oxytocin on brain activity in response to heat pain. Speculatively, oxytocin might influence emotional aspects of pain perception that have not been captured by our study as, for example, it has been shown that oxytocin enhances the pain-relieving effects of social support [46] and affects neural activity while seeing pain in others [47].

Methods/results:

-The description of the power analysis is unclear. Please provide more details.

Response: We have adjusted the description of the power analysis on page 5. Now it reads as follows:

The sample size was calculated with software G*Power 3. The calculation was done on the basis of a pilot experiment on conditioning of cortisol responses performed in our lab, as the design of this pilot corresponded to the design of the present study. The effect size found in the pilot experiment was d = 0.527. It was shown that 33 participants per group were necessary to obtain a power of .95 at an alpha level of a = .05. The power analysis was aimed at the question of the possibility to condition oxytocin release and not on the fMRI part of the trial.

-Description of statistical analyses (line 203-207) is unclear. Please reframe.

Response: We have adjusted the paragraph about the statistical analysis to make it clearer (page 10):

To investigate whether there was significant conditioned oxytocin release, the conditioned oxytocin group was compared to the placebo group without adding the exogenous oxytocin group into the analysis (as extremely high oxytocin levels were expected in the exogenous oxytocin group which was d prior to the study in the study registration protocol). The comparison was done with three (for each evocation day separately) repeated measures analyses of covariance (ANCOVA) with baseline oxytocin levels as a covariate. Next, the oxytocin administration group was added to the analyses and the three groups were compared on salivary oxytocin levels after the spray administration with repeated measures analyses of covariance in which baseline oxytocin levels served as a covariate. 

To investigate whether exogenous or conditioned oxytocin had an effect on the arousal ratings given during the Faces task, arousal ratings of faces were compared between the groups with factorial 4 (face valence: neutral, happy, angry or fearful) x 3 (group: oxytocin administration, placebo, conditioned oxytocin) ANOVA. 

Finally, an ANOVA was used to compare the groups on the temperatures that elicited a pain of 6 (on an 11-point NRS) that were used during the Pain task.

-Were there any group differences in baseline OT before the acquisition phase?

Response: We thank the reviewer for this question. There was no difference between the groups in their salivary oxytocin levels that were measured during the screening before the first acquisition day. We have added these results to the manuscript on page 12 and to Table 1:

There were no significant differences between the three groups on baseline salivary oxytocin levels on the screening (F (2, 80) = 1.01, p = .369)... 

-Grammatical error line 261 – 262: “was found” should be “were found”.

Response: we have corrected the typo.

-Were there any effects of conditioned oxytocin or intranasal oxytocin on the stimulus ratings (faces and cry sounds)?

Response: No differences were found between the groups in the arousal ratings. We have added the description of the analysis and results of this task to the methods on page 10 and results on page 15. Now it reads as follows:

To investigate whether exogenous or conditioned oxytocin had an effect on the arousal ratings given during the Faces task, arousal ratings of faces were compared between the groups with a factorial 4 (face valence: neutral, happy, angry or fearful) x 3 (group: oxytocin administration, placebo, conditioned oxytocin) ANOVA.

No differences were found between the groups in the arousal ratings given during the task (F (2, 77) = 0.15, p = .858). There was a significant difference in how arousing participants found faces of different modality (F (3, 77) = 116.85, p < .001). Bonferroni corrections demonstrated there were significant differences between all couples of modalities (all p’s < .05) and that happy faces were found to be the most arousing (M = 3.00), followed by fearful faces (M = 2.75), and angry faces (M = 2.5). Neutral faces (M = 1.23) were rated as the least arousing.

Discussion:

The authors should be more careful with interpreting insignificant results throughout the manuscript. E.g. line 444-447 � the authors mention that “effects remained small”, but effects were non-significant.

Response: We thank the reviewer for this remark. We certainly do not want to overstate the results of our study and therefore, we adjusted the sentence mentioned by the reviewer and also made some changes in the Abstract to emphasize that no group differences were found. Page 34:

Moreover, these effects were not generalizable across the tasks.

Abstract: 

The activation in the conditioned oxytocin group was in between the other two groups for these clusters but did not significantly differ from either group. No group differences were found in the other tasks. Preliminary evidence was found for brain activation of a conditioned oxytocin response; however, despite this trend in the expected direction, the conditioned group did not significantly differ from other groups. Future research should, therefore, investigate the optimal timing of conditioned endocrine responses and study whether the findings generalize to other hormones as well.

The authors found that intranasal oxytocin reduced STG activity, whereas previous studies point to enhanced STG activity. The authors should explain this discrepancy more clearly and discuss how their findings relate to previous research.

Response: We thank the reviewer for the opportunity to elaborate on this issue further. Indeed, we found that oxytocin reduced STG activity in response to fearful faces, which contradicts the generally found tendency of oxytocin to enhance the activation in STG (see meta-analyses: Grace et al., 2018, Psychoneuroendocrinology; Wang et al., 2017, Social Cognitive and Affective Neuroscience). At the same time, there is also some literature showing that oxytocin might decrease the activity in STG. Hech et al. (2017, Neuroimage) found that oxytocin reduces brain activation to social stimuli and particularly, that individuals with higher levels of social processing exhibited oxytocin induced decrease in STG in response to social stimuli. Also, a decrease in the activity of STG was found in response to social rejection (Gozzi et al., 2017, Neuropsychopharmacology). Our results add to the existing conflicting evidence regarding the effects of oxytocin on STG. 

We have extended the paragraph where we discuss these findings on page 30:

Moreover, the same fearful > neutral contrast yielded a significant difference between the oxytocin administration and placebo group in two clusters of the superior temporal gyrus (STG). The STG plays an important role in the processing of emotional stimuli and social cognition [40] and particularly processing of fearful faces [12, 41]. With this finding we thus replicated previous results showing increased STG activity in response to the presentation of the fearful faces. However, the direction of this oxytocin effect does not correspond to most previous studies. Several previous studies showed enhanced STG activity in response to emotional and social stimuli after oxytocin administration [26, 41]. However, in our study we found that participants in the oxytocin group had lower activation in the STG on the contrast fear > neutral in comparison to the placebo group. Some other studies also found dampening effects of oxytocin on STG activation. For example, a decrease in STG activity after oxytocin administration was found in response to social rejection [42]. Also, Hech and colleagues [43] demonstrated that oxytocin reduced brain activation to social stimuli and, particularly, that individuals with higher levels of social processing exhibited oxytocin induced decrease in STG in response to social stimuli. The findings on the directionality of STG brain activity in response to oxytocin are thus mixed in the current literature. In our study, we found an increase in STG in response to fearful faces in the placebo condition and this increase was dampened by oxytocin in the oxytocin condition, corresponding to our findings in the amygdala. Again, STG activity in the conditioned oxytocin group was in between the oxytocin and placebo groups but did not significantly differ from both groups. Possibly, similar to the results of the study on social rejection [42], oxytocin inhibited the processing of negative emotions of fearful faces in our study.

In the discussion, the authors could elaborate more on differential effects of exogenous oxytocin and manipulated endogenous oxytocin on brain activity. The increase in oxytocin induced by the conditioning paradigm is very small compared to the enormous increase induced by intranasal administration (Table 1). The authors do not find significant differences between the placebo group and conditioned oxytocin group. I wonder whether this may be due to the small increase in oxytocin, which may be insufficient to result in detectable brain activity changes.

Response: We thank the reviewer for this feedback. We have added a discussion of this issue to page 33 of the discussion:

Another possible explanation of the non-significant effects of conditioned oxytocin release on brain activity, is the difference in the magnitude of the effects between exogenous oxytocin administration and the conditioned response. Our data shows that even during evocation day 1, when the largest conditioned response was found in saliva, the oxytocin levels increased twice from the baseline, compared to a 100-time increase in the oxytocin administration group. It is unknown whether salivary oxytocin increase directly corresponds to the change in the brain activity, but it can be expected that the neural effects of conditioned oxytocin release might be much smaller than the effects of oxytocin administration. However, even small natural variations of the endogenous oxytocin levels have been shown to affect brain activity, for example, in resting state [50], during massage [51], and in response to aversive stimuli [52]. Therefore, it can be expected that endogenous oxytocin release triggered by conditioning, could be strong enough to affect brain activity.

6. PLOS authors have the option to publish the peer review history of their article (what does this mean?). If published, this will include your full peer review and any attached files.

Do you want your identity to be public for this peer review? For information about this choice, including consent withdrawal, please see our Privacy Policy.

Reviewer #1: No

Reviewer #2: Yes: Bart Boets

Reviewer #3: No

---

## [Decision Letter · Decision Letter 1]

12 Feb 2020

Effects of oxytocin administration and conditioned oxytocin on brain activity: an fMRI study.

PONE-D-19-27410R1

Dear Dr. Skvortsova,

We are pleased to inform you that your manuscript has been judged scientifically suitable for publication and will be formally accepted for publication once it complies with all outstanding technical requirements.

With kind regards,

Peter A. Bos

Academic Editor

PLOS ONE

Additional Editor Comments (optional):

Reviewers' comments:

Reviewer's Responses to Questions

**Comments to the Author**

1. If the authors have adequately addressed your comments raised in a previous round of review and you feel that this manuscript is now acceptable for publication, you may indicate that here to bypass the “Comments to the Author” section, enter your conflict of interest statement in the “Confidential to Editor” section, and submit your "Accept" recommendation.

Reviewer #1: All comments have been addressed

Reviewer #2: All comments have been addressed

Reviewer #3: All comments have been addressed

2. Is the manuscript technically sound, and do the data support the conclusions?

Reviewer #1: Yes

Reviewer #2: (No Response)

Reviewer #3: Yes

3. Has the statistical analysis been performed appropriately and rigorously? 

Reviewer #1: Yes

Reviewer #2: (No Response)

Reviewer #3: Yes

4. Have the authors made all data underlying the findings in their manuscript fully available?

Reviewer #1: Yes

Reviewer #2: (No Response)

Reviewer #3: Yes

5. Is the manuscript presented in an intelligible fashion and written in standard English?

Reviewer #1: Yes

Reviewer #2: (No Response)

Reviewer #3: Yes

6. Review Comments to the Author

Reviewer #1: The authors have adequately addressed all of my concerns. I can now recommend the article for publication.

Reviewer #2: (No Response)

Reviewer #3: (No Response)

7. PLOS authors have the option to publish the peer review history of their article (what does this mean?). If published, this will include your full peer review and any attached files.

Reviewer #1: Yes: tanya l procyshyn

Reviewer #2: No

Reviewer #3: No

---

## [Editor Report · Acceptance letter]

4 Mar 2020

PONE-D-19-27410R1 

Effects of oxytocin administration and conditioned oxytocin on brain activity: an fMRI study. 

Dear Dr. Skvortsova:

I am pleased to inform you that your manuscript has been deemed suitable for publication in PLOS ONE. Congratulations! Your manuscript is now with our production department. 

With kind regards,

on behalf of

Dr. Peter A. Bos 

Academic Editor

PLOS ONE